# Multiple-Attribute Text Rewriting

**Guillaume Lample**[*1,3], **Sandeep Subramanian**[*1,2], **Eric Michael Smith**[1],
**Ludovic Denoyer**[1,3], **Marc'Aurelio Ranzato**[1], **Y-Lan Boureau**[1]
[1]Facebook AI Research, [2]MILA, Université de Montréal
[3]Sorbonne Universités, UPMC Univ Paris 06
`sandeep.subramanian.1@umontreal.ca`
`{glample,ems,denoyer,ranzato,ylan}@fb.com`

## Abstract

The dominant approach to unsupervised "style transfer" in text is based on the idea of learning a latent representation, which is independent of the attributes specifying its "style". In this paper, we show that this condition is not necessary and is not always met in practice, even with domain adversarial training that explicitly aims at learning such disentangled representations. We thus propose a new model that controls several factors of variation in textual data where this condition on disentanglement is replaced with a simpler mechanism based on back-translation. Our method allows control over multiple attributes, like gender, sentiment, product type, etc., and a more fine-grained control on the trade-off between content preservation and change of style with a pooling operator in the latent space. Our experiments demonstrate that the fully entangled model produces better generations, even when tested on new and more challenging benchmarks comprising reviews with multiple sentences and multiple attributes.

## 1 Introduction

One of the objectives of unsupervised learning is to learn representations of data that enable fine control over the underlying latent factors of variation, e.g., pose and viewpoint of objects in images, or writer style and sentiment of a product review. In conditional generative modeling, these latent factors are given (Sohn et al., 2015; Mirza & Osindero, 2014; Ficler & Goldberg, 2017), or automatically inferred via observation of samples from the data distribution (Chen et al., 2017; 2016; Higgins et al., 2017).

More recently, several studies have focused on learning unsupervised mappings between two data domains such as images (Taigman et al., 2016; Isola et al., 2017; Zhu et al., 2017), words or sentences from different languages (Conneau et al., 2017; Lample et al., 2018).

In this problem setting, the generative model is conditioned not only on the desired attribute values, but also on a initial input, which it must transform. Generations should retain as many of the original input characteristics as possible, provided the attribute constraint is not violated. This learning task is typically unsupervised because no example of an input and its corresponding output with the specified attribute is available during training. The model only sees random examples and their attribute values.

The dominant approach to learn such a mapping in text is via an explicit constraint on disentanglement (Hu et al., 2017; Fu et al., 2017; Shen et al., 2017): the learned representation should be invariant to the specified attribute, and retain only attribute-agnostic information about the "content". Changing the style of an input at test time then amounts to generating an output based on the disentangled latent representation computed from the input and the desired attributes. Disentanglement is often achieved through an adversarial term in the training objective that aims at making the attribute value unrecoverable from the latent representation.

This paper aims to extend previous studies on "style transfer" along three axes. (i) First, we seek to gain a better understanding of what is necessary to make things work, and in particular, whether

---

[*]Authors contributed equally to this work

| Relaxed ↔ Annoyed | |
|---|---|
| Relaxed | Sitting by the Christmas tree and watching Star Wars after cooking dinner. What a nice night 💗 🎄 ✨ |
| Annoyed | Sitting by the computer and watching The Voice for the second time tonight. What a horrible way to start the weekend 😡 😡 😡 |
| Annoyed | Getting a speeding ticket 50 feet in front of work is not how I wanted to start this month 😒 |
| Relaxed | Getting a haircut followed by a cold foot massage in the morning is how I wanted to start this month 😊 |
| **Male ↔ Female** | |
| Male | Gotta say that beard makes you look like a Viking... |
| Female | Gotta say that hair makes you look like a Mermaid... |
| Female | Awww he's so gorgeous 😍 can't wait for a cuddle. Well done 😘 xxx |
| Male | Bro he's so f***ing dope can't wait for a cuddle. Well done bro |
| **Age 18-24 ↔ 65+** | |
| 18-24 | You cheated on me but now I know nothing about loyalty 😂 ok |
| 65+ | You cheated on America but now I know nothing about patriotism. So ok. |
| 65+ | Ah! Sweet photo of the sisters. So happy to see them together today . |
| 18-24 | Ah 😄 Thankyou 💕 #sisters 💗 happy to see them together today |

Table 1: Our approach can be applied to many different domains beyond sentiment flipping, as illustrated here with example re-writes by our model on public social media content. The first line in each box is an input provided to the model with the original attribute, followed by its rewrite when given a different attribute value.

disentanglement is key, or even actually achieved by an adversarial loss in practice. In Sec. 3.1 we provide strong empirical evidence that disentanglement is not necessary to enable control over the factors of variation, and that even a method using adversarial loss to disentangle (Fu et al., 2017) does not actually learn representations that are disentangled. (ii) Second, we introduce a model which replaces the adversarial term with a back-translation (Sennrich et al., 2015a) objective which exposes the model to a pseudo-supervised setting, where the model's outputs act as supervised training data for the ultimate task at hand. The resulting model is similar to recently proposed methods for unsupervised machine translation (Lample et al., 2017a; 2018; Artetxe et al., 2018; Zhang et al., 2018b), but with two major differences: (a) we use a pooling operator which is used to control the trade-off between style transfer and content preservation; and (b) we extend this model to support multiple attribute control. (iii) Finally, in Sec. 4.1 we point out that current style transfer benchmarks based on collections of user reviews have severe limitations, as they only consider a single attribute control (sentiment), and very small sentences in isolation with noisy labels. To address this issue, we propose a new set of benchmarks based on existing review datasets, which comprise full reviews, where multiple attributes are extracted from each review.

The contributions of this paper are thus: (1) a deeper understanding of the necessary components of style transfer through extensive experiments, resulting in (2) a generic and simple learning framework based on mixing a denoising auto-encoding loss with an online back-translation technique and a novel neural architecture combining a pooling operator and support for multiple attributes, and (3) a new, more challenging and realistic version of existing benchmarks which uses full reviews and multiple attributes per review, as well as a comparison of our approach w.r.t. baselines using both new metrics and human evaluations. We will open-source our code and release the new benchmark datasets used in this work, as well as our pre-trained classifiers and language models for reproducibility. This will also enable fair empirical comparisons on automatic evaluation metrics in future work on this problem.

## 2 RELATED WORK

There is substantial literature on the task of unsupervised image translation. While initial approaches required supervised data of the form *(input, transformation, output)*, e.g., different images of the same object rendered with different viewpoints or/and different lighting conditions (Hinton et al., 2011; Yang et al., 2015; Kulkarni et al., 2015), current techniques are capable of learning completely unsupervised domain mappings. Given images from two different domains $\mathcal{X}$ and $\mathcal{Y}$ (where $\mathcal{X}$ could be the domain of paintings and $\mathcal{Y}$ the domain of realistic photographs), and the task is to learn two mappings $F : \mathcal{X} \rightarrow \mathcal{Y}$ and $G : \mathcal{Y} \rightarrow \mathcal{X}$, without supervision, i.e., just based on images sampled from the two domains (Liu & Tuzel, 2016; Taigman et al., 2016; Isola et al., 2017). For instance, Zhu et al. (2017) used a cycle consistency loss to enforce $F(G(y)) \approx y$ and $G(F(x)) \approx x$. This loss is minimized along with an adversarial loss on the generated outputs to constrain the model to generate realistic images. In Fader Networks (Lample et al., 2017b), a discriminator is applied on

the latent representation of an image autoencoder to remove the information about specific attributes. The attribute values are instead given explicitly to the decoder at training time, and can be tuned at inference to generate different realistic versions of an input image with varying attribute values.

Different approaches have been proposed for textual data, mainly aiming at controlling the writing style of sentences. Unfortunately, datasets of parallel sentences written in a different style are hard to come by. Carlson et al. (2017) collected a dataset of 33 English versions of the Bible written in different styles on which they trained a supervised style transfer model. Li et al. (2018) released a small crowdsourced subset of 1,000 Yelp reviews for evaluation purposes, where the sentiment had been swapped (between *positive* and *negative*) while preserving the content. Controlled text generation from unsupervised data is thus the focus of more and more research.

An theme that is common to most recent studies is that style transfer can be achieved by disentangling sentence representations in a shared latent space. Most solutions use an adversarial approach to learn latent representations agnostic to the style of input sentences (Fu et al., 2017; Hu et al., 2017; Shen et al., 2017; Zhang et al., 2018a; Xu et al., 2018; John et al., 2018; Zhao et al., 2018). A decoder is then fed with the latent representation along with attribute labels to generate a variation of the input sentence with different attributes.

Unfortunately, the discrete nature of the sentence generation process makes it difficult to apply to text techniques such as cycle consistency or adversarial training. For instance, the latter (Shen et al., 2017; dos Santos et al., 2018; Zhang et al., 2018c) requires methods such as REINFORCE (He et al., 2016) or approximating the output softmax layer with a tunable temperature (Hu et al., 2017; Prabhumoye et al., 2018; Yang et al., 2018), all of which tend to be slow, unstable and hard to tune in practice. Moreover, all these studies control a single attribute (e.g. swapping positive and negative sentiment).

The most relevant work to ours is Zhang et al. (2018b), which also builds on recent advances in unsupervised machine translation. Their approach first consists of learning cross-domain word embeddings in order to build an initial phrase-table. They use this phrase-table to bootstrap an iterative back-translation pipeline containing both phrase-based and neural machine translation systems. Overall, their approach is significantly more complicated than ours, which is end-to-end and does not require any pre-training. Moreover, this iterative back-translation approach has been shown to be less effective than on-the-fly back-translation which is end-to-end trainable (Lample et al., 2018).

## 3    CONTROLLABLE TEXT REWRITING

This section briefly introduces notation, the task, and our empirical procedure for evaluating disentanglement before presenting our approach.

We consider a training set $\mathcal{D} = (x^i, y^i)_{i \in [1,n]}$ of $n$ sentences $x^i \in \mathcal{X}$ paired with attribute values $y^i$. $y \in \mathcal{Y}$ is a set of $m$ attribute values $y = (y_1, ..., y_m)$. Each attribute value $y_k$ is a discrete value in the set $\mathcal{Y}_k$ of possible values for attribute $k$, e.g. $\mathcal{Y}_k = \{\text{bad}, \text{neutral}, \text{good}\}$ if $y_k$ represents the overall rating of a restaurant review.

Our task is to learn a model $F : \mathcal{X} \times \mathcal{Y} \to \mathcal{X}$ that maps any pair $(x, \tilde{y})$ of an input sentence $x$ (whose actual set of attributes are $y$) and a new set of $m$ attribute values $\tilde{y}$ to a new sentence $\tilde{x}$ that has the specified attribute values $\tilde{y}$, subject to retaining as much as possible of the original content from $x$, where content is defined as anything in $x$ which does not depend on the attributes.

The architecture we consider performs this mapping through a sequence-to-sequence auto-encoder that first encodes $x$ into a latent representation $z = e(x)$, then decodes $(z, \tilde{y})$ into $\tilde{x} = d(z, \tilde{y})$, where $e$ and $d$ are functions parameterized by the vector of trainable parameters $\theta$. Before giving more detail on the architecture, let us look at disentanglement.

### 3.1    ARE ADVERSARIAL MODELS REALLY DOING DISENTANGLEMENT?

Almost all the existing methods are based on the common idea to learn a latent representation $z$ that is disentangled from $y$. We consider $z$ to be disentangled from $y$ if it is impossible to recover $y$ from $z$. While failure to recover $y$ from $z$ could mean either that $z$ was disentangled or that the classifier

chosen to recover $y$ was either not powerful enough or poorly trained, success of *any* classifier in recovering $y$ demonstrates that $z$ was in fact not invariant to $y$.

| $\lambda_{adv}$ | Discriminator Acc (Train) | Post-fit Classifier Acc (Test) |
|---|---|---|
| 0 | 89.45% | 93.8% |
| 0.001 | 85.04% | 92.6% |
| 0.01 | 75.47% | 91.3% |
| 0.03 | 61.16% | 93.5% |
| 0.1 | 57.63% | 94.5% |
| 1.0 | 52.75% | 86.1% |
| 10 | 51.89% | 85.2% |
| fastText | - | 97.7% |

Table 2: Recovering the sentiment of the input from the encoder's representations of a domain adversarially-trained Fader model (Fu et al., 2017). During training, the discriminator, which was trained adversarially and jointly with the model, gets worse at predicting the sentiment of the input when the coefficient of the adversarial loss $\lambda_{adv}$ increases. However, a classifier that is separately trained on the resulting encoder representations has an easy time recovering the sentiment. We also report the baseline accuracy of a fastText classifier trained on the actual inputs.

As a preliminary study, we gauge the degree of disentanglement of the latent representation. Table 2 shows that the value of the attribute can be well recovered from the latent representation of a Fader-like (Fu et al., 2017) model even when the model is trained adversarially. A classifier fit post-hoc and trained from scratch, parameterized identically to the discriminator(see paragraph on model architecture in Section 3.3 for details), is able to recover attribute information from the "distengeled" content representation learned via adversarial training. This suggests that disentanglement may not be achieved in practice, even though the discriminator is unable to recover attribute information well during training. We do not assert that disentangled representations are undesirable but simply that it isn't mandatory in the goal of controllable text rewriting. This is our focus in the following sections.

## 3.2 OUR APPROACH

Evaluation of controlled text generation can inform the design of a more streamlined approach: generated sentences should (1) be fluent, (2) make use of the specified attribute values, and (3) preserve the rest of the content of the input.

Denoising auto-encoding (DAE) (Fu et al., 2017) is a natural way to learn a generator that is both fluent and that can reconstruct the input, both the content and the attributes. Moreover, DAE is a weak way to learn about how to change the style, or in other words, it is a way to force the decoder to also leverage the externally provided attribute information. Since the noise applied to the encoder input $x$ may corrupt words conveying the values of the input attribute $y$, the decoder has to learn to use the additional attribute input values in order to perform a better reconstruction. We use the noise function described in Lample et al. (2017a) that corrupts the input sentence by performing word drops and word order shuffling. We denote by $x_c$ a corrupted version of the sentence $x$.

As discussed in Sec. 3.1, disentanglement is not necessary nor easily achievable, and therefore, we do not seek disentanglement and do not include any adversarial term in the loss. Instead, we consider a more natural constraint which encourages the model to perform well at the task we are ultimately interested in - controlled generation via externally provided attributes. We take an input $(x, y)$ and encode $x$ it into $z$, but then decode using another set of attribute values, $\tilde{y}$, yielding the reconstruction $\tilde{x}$. We now use $\tilde{x}$ as input of the encoder and decode it using the original $y$ to ideally obtain the original $x$, and we train the model to map $(\tilde{x}, y)$ into $x$. This technique, called back-translation (BT) (Sennrich et al., 2015a; Lample et al., 2017a; 2018; Artetxe et al., 2018), has a two-fold benefit. Initially when the DAE is not well trained and $\tilde{x}$ has lost most of the content present in $x$, the only useful information provided to the decoder is the desired attribute $y$. This encourages the decoder to leverage the provided attributes. Later on during training when DAE is better, BT helps training the sequence-to-sequence for the desired task. Overall, we minimize:

$$\mathcal{L} = \lambda_{AE} \sum_{(x,y)\sim\mathcal{D}} -\log p_d\Big(x|e(x_c), y\Big) + \lambda_{BT} \sum_{(x,y)\sim\mathcal{D}, \tilde{y}\sim\mathcal{Y}} -\log p_d\Big(x|e\big(d(e(x), \tilde{y})\big), y\Big) \quad (1)$$

where $p_d$ is the probability distribution over sequences $x$ induced by the decoder, $e(x_c)$ is the encoder output when fed with a corrupted version $x_c$ of the input $x$, and $d(e(x), \tilde{y})$ is a variation of the input sentence $x$ written with a randomly sampled set of attributes $\tilde{y}$. In practice, we generate sentences during back-translation by sampling from the multinomial distribution over words defined by the decoder at each time step using a temperature $T$.

### 3.3 IMPLEMENTATION

So far, the model is the same as the model used for unsupervised machine translation by Lample et al. (2018), albeit with a different interpretation of its inner workings, no longer based on disentanglement. Instead, the latent representation $z$ can very well be entangled, but we only require the decoder to eventually "overwrite" the original attribute information with the desired attributes. Unfortunately, this system may be limited to swapping a single binary attribute and may not give us enough control on the trade-off between content preservation and change of attributes. To address this limitations, we introduce the following components:

**Attribute conditioning** In order to handle multiple attributes, we separately embed each target attribute value and then average their embeddings. We then feed the averaged embeddings to the decoder as a start-of-sequence symbol.

We also tried an approach similar to Michel & Neubig (2018), where the output layer of the decoder uses a different bias for each attribute label. We observed that the learned biases tend to reflect the labels of the attributes they represent. Examples of learned biases can be found in Table 14. However, this approach alone did not work as well as using attribute-specific start symbols, nor did it improve results when combined with them.

**Latent representation pooling** To control the amount of content preservation, we use pooling. The motivating observation is that models that compute one latent vector representation per input word usually perform individual word replacement, while models without attention are much less literal and tend to lose content but have an easier time changing the input sentence with the desired set of attributes. Therefore, we propose to gain finer control by adding a temporal max-pooling layer on top of the encoder, with non-overlapping windows of width $w$. Setting $w = 1$ results in a standard model with attention, while setting $w$ to the length of the input sequence boils down to a sequence-to-sequence model without attention. Intermediate values of $w$ allow for different trade-offs between preserving information about the input sentence and making the decoder less prone to copying words one by one.

The hyper-parameters of our model are: $\lambda_{AE}$ and $\lambda_{BT}$ trading off the denoising auto-encoder term versus the back-translation term (the smaller the $\lambda_{BT}/\lambda_{AE}$ ratio the more the content is preserved and the less well the attributes are swapped), the temperature $T$ used to produce unbiased generations (Edunov et al., 2018) and to control the amount of content preservation, and the pooling window size $w$. We optimize this loss by stochastic gradient descent without back-propagating through the back-translation generation process; back-translated sentences are generated on-the-fly once a new mini-batch arrives.

**Model Architecture** We use an encoder parameterized by a 2-layer bidirectional LSTM and a 2-layer decoder LSTM augmented with an attention mechanism (Bahdanau et al., 2014). Both LSTMs and our word embedding lookup tables, trained from scratch, have 512 hidden units. Another embedding lookup table with 512 hidden units is used to embed each attribute value. The decoder conditions on two different sources of information: 1) attribute embedding information that presented that it as the first token, similar to Lample et al. (2018) and at the softmax output as an attribute conditional bias following Michel & Neubig (2018). When controlling multiple attributes, we average the embeddings and bias vectors that correspond to the different attribute values. 2) The decoder also conditions on a temporally downsampled representation of the encoder via an attention mechanism. The representations are downsampled by temporal max-pooling with a non-overlapping window of size 5. Although our best models do not use adversarial training, in ablations and experiments that study disentanglement, we used a discriminator paramaeterized as 3 layer MLP with 128 hidden units and LeakyReLU acivations.

## 4 EXPERIMENTS

### 4.1 DATASETS

We use data from publicly available Yelp restaurant and Amazon product reviews following previous work in the area (Shen et al., 2017; Li et al., 2018) and build on them in three ways to make the task more challenging and realistic. Firstly, while previous approaches operate at the sentence level by assuming that every sentence of a review carries the same sentiment as the whole of review, we operate at the granularity of entire reviews. The sentiment, gender[1] of the author and product/restaurant labels are therefore more reliable. Secondly, we relax constraints enforced in prior works that discard reviews with more than 15 words and only consider the 10k most frequent words. In our case, we consider full reviews with up to 100 words, and we consider byte-pair encodings (BPE) Sennrich et al. (2015b) with 60k BPE codes, eliminating the presence of unknown words. Finally, we leverage available meta-data about restaurant and product categories to collect annotations for two additional controllable factors: the gender of the review author and the category of the product or restaurant being reviewed. A small overview of the corresponding datasets is presented below with some statistics presented in Table 3. Following Li et al. (2018), we also collect human reference edits for sentiment and restaurant/product categories to serve as a reference for automatic metrics as well as an upper bound on human evaluations (examples in Appendix Table 12).

**Yelp Reviews** This dataset consists of restaurant and business reviews provided by the Yelp Dataset Challenge[2]. We pre-process this data to remove reviews that are either 1) not written in English according to a fastText (Joulin et al., 2016) classifier, 2) not about restaurants, 3) rated 3/5 stars as they tend to be neutral in sentiment (following Shen et al. (2017)), or 4) where the gender is not identifiable by the same method as in Reddy & Knight (2016); Prabhumoye et al. (2018). We then binarize both sentiment and gender labels. Five coarse-grained restaurant category labels, *Asian, American, Mexican, Bars & Dessert*, are obtained from the associated meta-data. Since a review can be written about a restaurant that has multiple categories (ex: an Asian restaurant that serves desserts), we train a multi-label fastText classifier to the original data that has multiple labels per example. We then re-label the entire dataset with this classifier to pick the most likely category to be able to model the category factor as a categorical random variable. (See Appendix section A.2 for more details.) Since there now exists two variants of the Yelp dataset, we refer to the one used by previous work (Shen et al., 2017; Fu et al., 2017; Li et al., 2018) as *SYelp* and our created version with full reviews along with gender and category information as *FYelp* henceforth.

**Amazon Reviews** The amazon product review dataset (He & McAuley, 2016) is comprised of reviews written by consumers of Amazon products. We followed the same pre-processing steps as in the Yelp dataset with the exception of collecting gender labels, since a very large fraction of amazon usernames were not present in a list of gender-annotated names. We labeled reviews with the following product categories based on the meta-data: *Books*, *Clothing*, *Electronics*, *Movies*, *Music*. We followed the same protocol as in *FYelp* to re-label product categories. In this work, we do not experiment with the version of the Amazon dataset used by previous work, and so we refer to our created version with full reviews along with product category information as just *Amazon* henceforth. Statistics about the dataset can be found in Table 3.

**Public social media content** We also used an unreleased dataset of public social media content written by English speakers to illustrate the approach with examples from a more diverse set of categories[3]. We used 3 independent pieces of available information about that content: 1) gender (male or female) 2) age group (18-24 or 65+), and 3) writer-annotated feeling (relaxed or annoyed).

---

[1]Note that using "gender" (or any other attribute for that matter) as a differentiating attribute between several bodies of text implies that there are indeed signatures of gender in the data. These signatures could be as innocuous as some first names like Mary being usually associated with women, or disheartening like biases and stereotypes exposed by statistical methods, (e.g., "man is to computer programmer as woman is to homemaker" (Bolukbasi et al., 2016)). We certainly do not condone those stereotypes, and on the contrary, we hope that showing that our models can uncover these biases might down the line turn them into powerful tools for researchers who study fairness and debiasing (Reddy & Knight, 2016).

[2]https://www.yelp.com/dataset/challenge

[3]See footnote 1.

| | Sentiment | | Gender | | Category | | | | |
|---|---|---|---|---|---|---|---|---|---|
| **SYelp** | Positive | Negative | Male | Female | American | Asian | Bar | Dessert | Mexican |
| | 266,041 | 177,218 | - | - | - | - | - | - | - |
| **FYelp** | Positive | Negative | Male | Female | American | Asian | Bar | Dessert | Mexican |
| | 2,056,132 | 639,272 | 1,218,068 | 1,477,336 | 904,026 | 518,370 | 595,681 | 431,225 | 246,102 |
| **Amazon** | Positive | Negative | - | - | Book | Clothing | Electronics | Movies | Music |
| | 64,251,073 | 10,944,310 | - | - | 26,208,872 | 14,192,554 | 25,894,877 | 4,324,913 | 4,574,167 |
| **Social Media Content** | Relaxed | Annoyed | Male | Female | 18-24 | 65+ | | | |
| | 7,682,688 | 17,823,468 | 14,501,958 | 18,463,789 | 12,628,250 | 7,629,505 | | | |

Table 3: The number of reviews for each attribute for different datasets. The *SYelp*, *FYelp* and the Amazon datasets are composed of 443k, 2.7M and 75.2M sentences respectively. Public social media content is collected from 3 different data sources with 25.5M, 33.0M and 20.2M sentences for the *Feeling*, *Gender* and *Age* attributes respectively.

To make the data less noisy, we trained a fastText classifier (Joulin et al., 2016) for each attribute and only kept the data above a certain confidence threshold.

## 4.2 EVALUATION

Automatic evaluation of generative models of text is still an open research problem. In this work, we use a combination of multiple automatic evaluation criteria informed by our desiderata. We would like our systems to *simultaneously* 1) produce sentences that conform to the set of pre-specified attribute(s), 2) preserve the structure and content of the input, and 3) generate fluent language. We therefore evaluate samples from different models along three different dimensions:

- **Attribute control:** We measure the extent to which attributes are controlled using fastText classifiers, trained on our datasets, to predict different attributes.

- **Fluency:** Fluency is measured by the perplexity assigned to generated text sequences by a pre-trained Kneser–Ney smooth 5-gram language model using KenLM (Heafield, 2011).

- **Content preservation:** We measure the extent to which a model preserves the content present of a given input using n-gram statistics, by measuring the *BLEU* score between generated text and the input itself, which we refer to as *self-BLEU*. When a human reference is provided instead, we compute the BLEU score with respect to it, instead of the input, which we will refer to as just *BLEU* (Papineni et al., 2002). "BLEU" scores in this paper correspond to the BLEU score with respect to human references *averaged* across generations conditioned on all possible attribute values *except for that of the input*. However, when reporting self-BLEU scores, we also average across cases where generations are also conditioned on the same attribute value as the input.

A combination of these metrics, however, only provides a rough understanding of the quality of a particular model. Ultimately, we rely on human evaluations collected via a public crowd-sourcing platform. We carried out two types of evaluations to compare different models. 1) Following a protocol similar to Li et al. (2018), we ask crowd workers to annotate generated sentences along the three dimensions above. Fluency and content preservation are measured on a likert-scale from 1 to 5 and attribute control is evaluated by asking the worker to predict the attribute present in the generated text. 2) We take a pair of generations from two different models, and ask workers to pick the generation they prefer on the overall task, accounting for all the dimensions simultaneously. They are also presented with a "no preference" option to discard equally good or bad generations from both models.

## 4.3 MODEL SELECTION

Since our automatic evaluation metrics are only weak proxies for the quality of a model, we set minimum thresholds on the content preservation and attribute control criteria and only consider models above a certain threshold.

The few models that met the specified threshold on the validation set were evaluated by humans on the same validation set and the best model was selected to be run on the test set.

| Model | Accuracy | BLEU | PPL |
|---|---|---|---|
| Fader/StyleEmbedding (Fu et al., 2017) | 18% | 16.7 | 56.1 |
| MultiDecoder (Fu et al., 2017) | 52% | 11.3 | 90.1 |
| ControllableText (Hu et al., 2017) | 85% | 20.6 | 232.0 |
| CAE (Shen et al., 2017) | 72% | 6.8 | 53.0 |
| Retrieval (Li et al., 2018) | 81% | 1.3 | 7.4 |
| Rule-based (Li et al., 2018) | 73% | 22.3 | 118.7 |
| DeleteOnly (Li et al., 2018) | 77% | 14.5 | 67.1 |
| DeleteAndRetrieve (Li et al., 2018) | 79% | 16.0 | 66.6 |
| Fader (Ours w/o backtranslation & attention) | 71% | 15.7 | 35.1 |
| Ours | 87% | 14.6 | 26.2 |
| Ours | 85% | 24.2 | 26.5 |
| Ours | 74% | 31.2 | 49.8 |
| Input copy | 13% | 30.6 | 40.6 |

Table 4: Automatic evaluation of models on the *SYelp* test set from Li et al. (2018). The test set is composed of sentences that have been manually written by humans, which we use to compute the BLEU score. Samples for previous models were made available by Li et al. (2018). For our model, we report different results corresponding to different choices of hyper-parameters (pooling kernel width and back-translation temperature) to demonstrate our model's ability to control the trade-off between attribute transfer and content preservation.

## 4.4 Comparisons to Prior Work

Our first set of experiments aims at comparing our approach with different models recently proposed, on the *SYelp* dataset. Results using automatic metrics are presented in Table 4. We compare the same set of models as in Li et al. (2018) with the addition of our model and our own implementation of the Fader network (Lample et al., 2017b), which corresponds to our model without back-translation and without attention mechanism, but uses domain adversarial training (Ganin et al., 2016) to remove information about sentiment from the encoder's representation. This is also similar to the StyleEmbedding model presented by Fu et al. (2017). For our approach, we were able to control the trade-off between BLEU and accuracy based on different hyper-parameter choices. We demonstrate that our approach is able to outperform all previous approaches on the three desired criteria simultaneously, while our implementation of the fader is competitive with the previous best work.

| | Fluency | Content | Sentiment |
|---|---|---|---|
| DAR (Li et al. (2018)) | 3.33 (1.39) | 3.16 (1.43) | 64.05% |
| Ours | 4.07 (1.12) | 3.67 (1.41) | 69.66% |
| Human (Li et al. (2018)) | 4.56 (0.78) | 4.01 (1.25) | 81.35% |

| | Our Model | No Preference | DAR |
|---|---|---|---|
| DAR vs Our Fader | **37.6%** | 32.7% | 29.7% |
| DAR vs Ours | **54.4%** | 24.7% | 20.8% |

Table 5: **Top:** Results from human evaluation to evaluate the fluency / content preservation and successful sentiment control on the Li et al. (2018) *SYelp* test set. The mean and standard deviation of Fluency and Content are measured on a likert scale from 1-5 while sentiment is measured by fraction of times that the controlled sentiment of model matches the judge's evaluation of the sentiment (when also presented with a neutral option). **Bottom:** Results from human A/B testing of different pairs of models. Each cell indicates the fraction of times that a judge preferred one of the models or neither of them on the overall task.)

Since our automatic evaluation metrics are not ideal, we carried out human evaluations using the protocol described in Section 4.2. Table 5 (top) shows the fluency, content preservation and attribute control (sentiment) scores obtained by our model, DeleteAndRetrieve (DAR) and turkers from Li

et al. (2018) [4] on the *SYelp* dataset. While humans clearly outperform both models, our model is better than DeleteAndRetrieve on all 3 dimensions. We further demonstrate our model's strength over DeleteAndRetrieve in Table 5 (bottom) in an A/B test between two the models, where crowd workers prefer our model 54.4% compared to theirs 20.8%. Interestingly, our baseline Fader model is also able to do better (37.6% vs 29.7%), suggesting limitations in our automatic metrics, since Fader does not do as well in Table 4.

## 4.5 EVALUATING MULTIPLE ATTRIBUTE CONTROL

| Dataset (Model) | Attributes | Sentiment | | Category | | Gender | |
|---|---|---|---|---|---|---|---|
| | | Accuracy | self-BLEU | Accuracy | self-BLEU | Accuracy | self-BLEU |
| Yelp (DAR) | Sentiment | 78.7% | 42.1 | - | - | - | - |
| Yelp (Our Fader) | Sentiment | 85.5% | 31.3 | - | - | - | - |
| | Sentiment + Category | 85.1% | 20.6 | 46.1% | 22.6 | - | - |
| | Sentiment + Category + Gender | 86.6% | 20.4 | 47.7% | 22.5 | 58.5% | 23.3 |
| Yelp (Ours) | Sentiment | 87.4% | 54.5 | - | - | - | - |
| | Sentiment + Category | 87.1% | 38.8 | 64.9% | 44.0 | - | - |
| | Sentiment + Category + Gender | 88.5% | 31.6 | 64.1% | 36.5 | 59.0% | 37.4 |
| | Gender | - | - | - | - | 59.1% | 47.0 |
| Amazon (Ours) | Sentiment | 82.6% | 54.8 | - | - | - | - |
| | Sentiment + Category | 82.5% | 48.9 | 81.4% | 41.8 | - | - |
| Input Copy | - | 50.0% | 100.0 | 20.0% | 100.0 | 50.0% | 100.0 |

Table 6: Results using automatic evaluation metrics on the *FYelp* and *Amazon* test sets. Different rows correspond to the set of attributes being controlled by the model.

Table 6 presents the quantitative results obtained by the *FYelp* and *Amazon* datasets when controlling single and multiple attributes. For this table, unlike for Table 4, the DeleteAndRetrieve (DAR) results were obtained by re-training the model of Li et al. (2018). We use our implementation of the Fader model since we found it to be better than previous work, by human evaluation (Table 5). While we control all attributes simultaneously during training, at test time, for the sake of quantitative evaluations, we change the values only of a single attribute while keeping the others constant. Our model clearly outperforms the baseline Fader model. We also demonstrate that our model does not suffer significant drops in performance when controlling multiple attributes over a single one.

Demonstrations of our model's ability to control single and multiple attributes are presented in Table 8 and Table 9 respectively. What is interesting to observe is that our model does not just alter single words in the input to control an attribute, but often changes larger fragments to maintain grammaticality and fluency. Examples of re-writes by our model on social media content in Table 1 show that our model tends to retain the overall structure of input sentences, including punctuation and emojis. Additional examples of re-writes can be found in Table 10 and Table 11 in Appendix.

## 4.6 ABLATION STUDY

| Model | Test (FYelp) | | Test (Li et al., 2018) | |
|---|---|---|---|---|
| | Accuracy | self-BLEU | Accuracy | BLEU |
| Our model | 87% | 54.5 | 80% | 25.8 |
| -pooling | 89% | 47.9 | - | - |
| -temperature | 86% | 45.2 | 80% | 21.3 |
| -attention | 93% | 25.4 | 80% | 22.1 |
| -back-translation | 86% | 32.8 | 69% | 16.4 |
| +adversarial | 86% | 45.5 | 78% | 25.1 |
| -attention -back-translation | 90% | 26.0 | 71% | 15.7 |

Table 7: Model ablations on 5 model components on the *FYelp* dataset (Left) and *SYelp* (Right).

---

[4]https://github.com/lijuncen/Sentiment-and-Style-Transfer/tree/master/evaluation/outputs/yelp

In Table 7, we report results from an ablation study on the *SYelp* and *FYelp* datasets to understand the impact of the different model components on overall performance. The different components are: 1) pooling, 2) temperature based multinomial sampling when back-translating, 3) attention, 4) back-translation, 5) the use of domain adversarial training and 6) attention and back-translation in conjunction. We find that a model with all of these components, except for domain adversarial training, performs the best, further validating our hypothesis in Section 3.1 that it is possible to control attributes of text without disentangled representations. The absence of pooling or softmax temperature when back-translating also has a small negative impact on performance, while the attention and back-translation have much bigger impacts.

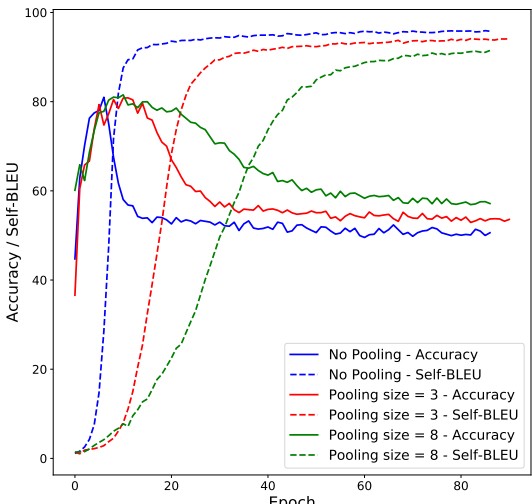

Figure 1: Accuracy and self-BLEU curves on the *FYelp* dataset for different pooling operator configurations. Without pooling, the model tends to converge to a copy mode very quickly, with a high self-BLEU score and a poor accuracy. The pooling operator alleviates this behaviour and provides models with a different trade-off accuracy / content preservation.

Table 13 shows examples of reviews re-written by different models at different checkpoints, showing the trade-off between properly modifying the attribute and preserving the original content. Figure 1 shows how the trade-off between content preservation (self-BLEU) and attribute control (accuracy) evolves over the course of training and as a function of the pooling kernel width.

## 5 CONCLUSION

We present a model that is capable of re-writing sentences conditioned on given attributes, that is not based on a disentanglement criterion as often used in the literature. We demonstrate our model's ability to generalize to a realistic setting of restaurant/product reviews consisting of several sentences per review. We also present model components that allow fine-grained control over the trade-off between attribute control versus preserving the content in the input. Experiments with automatic and human-based metrics show that our model significantly outperforms the current state of the art not only on existing datasets, but also on the large-scale datasets we created. The source code and benchmarks will be made available to the research community after the reviewing process.

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

# A    SUPPLEMENTARY MATERIAL

## A.1    TRAINING DETAILS

We used the Adam optimizer (Kingma & Ba, 2014) with a learning rate of $10^{-4}$, $\beta_1 = 0.5$, and a batch size of 32. As in Lample et al. (2018), we fix $\lambda_{BT} = 1$, and set $\lambda_{AE}$ to 1 at the beginning of the experiment, and linearly decrease it to 0 over the first $300,000$ iterations. We use greedy decoding at inference. When generating pseudo-parallel data via back-translation, we found that increasing the temperature over the course of training from greedy generation to multinomial sampling with a temperature of 0.5 linearly over 300,000 steps was useful (Edunov et al., 2018). Since the class distribution for different attributes in both the Yelp and Amazon datasets are skewed, we train with balanced minibatches when there is only a single attribute being controlled and with independent and uniformly sampled attribute values otherwise. The synthetic target attributes during back-translation $\tilde{y}$ are also balanced by uniform sampling.

## A.2    DATASET CREATION DETAILS

In addition to the details presented in Section 4.1, we present additional details on the creation of the *FYelp* and *Amazon* datasets.

**FYelp:**    Reviews, their rating, user information and restaurant/business categories are obtained from the available metadata. We construct sentiment labels by grouping 1/2 star ratings into the negative category and 4/5 into the positive category while discarding 3 star reviews. To determine the gender of the person writing a review, we obtain their name from the available user information and then look it up in a list of gendered names[5] following Prabhumoye et al. (2018); Reddy & Knight (2016). We discard reviews for which we were unable to obtain gender information with this technique. Restaurant/business category meta-data is available for each review, from which we discard all reviews that were not written about restaurants. Amongst restaurant reviews, we manually group restaurant categories into "parent" categories to cover a significant fraction of the dataset. The grouping is as follows:

- **Asian** - Japanese, Thai, Ramen, Sushi, Sushi Bar, Chinese, Asian Fusion, Vietnamese, Korean, Noodles, Dim Sum, Cantonese, Filipino, Taiwanese
- **American** - American (New), American (Traditional), Canadian (New), Southern
- **Mexican/Latin American** - New Mexican Cuisine, Mexican, Tacos, Tex-Mex, Tapas Bars, Latin American
- **Bars** - Brasseries, Nightlife, Bars, Pubs, Wine Bars, Sports Bars, Beer, Cocktail Bars
- **Desserts** - Desserts, Bakeries, Ice Cream & Frozen Yogurt, Juice Bars & Smoothies Donuts, Cupcakes, Chocolatiers & Shops

As described in Section 4.1, we train a classifier on these parent categories and relabel the entire dataset using this.

**Amazon:**    Reviews, their rating, user information and restaurant/business categories are obtained from the metadata made available by He & McAuley (2016). We construct sentiment labels in the same manner as in *FYelp*. We did not experiment with gender labels, since we found that Amazon usernames seldom use real names. We group Amazon product categories into "parent categories" manually, similar to *FYelp* as follows:

- **Books** - Books, Books & Comics, Children's Books, Literature & Fiction, Comic Books, Kindle eBooks etc.
- **Electronics** - Car Electronics, Cell Phones, Electrical & Electronics, Electronics, Electronics & Gadgets, Mobiles, Tablets, Headphones etc.
- **Movies** - Movies, Movies & TV, Movies & Video, TV & Film
- **Clothing** - Clothing, Shoes & Jewelry, Baby Clothing, Fashion

---

[5] https://www.ssa.gov/oact/babynames/names.zip

- **Music** - CDs & Vinyl, Music, Digital Music, Children's Music, World Music, Electronic Music

We relabel reviews with a trained product category classifier similar to *FYelp*.

For the *FYelp* and *Amazon* datasets, we normalize, lowercase and tokenize reviews using the moses (Koehn et al., 2007) tokenizer. With social media content, we do not lowercase data in order to exploit interesting capitalization patterns inherent in the data, but we still run other pre-processing steps. We use byte-pair encodings (BPE) (Sennrich et al., 2015b) with 60k replacements, on all 3 datasets, to deal with large vocabulary sizes.

**Human Annotated References** Li et al. (2018) released a set of human reference edits when controlling the sentiment of a review, on a test set of 500 examples on the *SYelp* dataset. We follow suit by collecting a similar dataset of 500 human reference edits, which will be made publicly available, for both sentiment and product categories on the *FYelp* and *Amazon* datasets. When collecting such data, we use pre-trained sentiment/category classifiers to interactively guide crowd workers using the ParlAI (Miller et al., 2017) platform, to produce edits with the desired attribute value as well as significant content overlap with the input.

## A.3 ADDITIONAL QUALITATIVE EXAMPLES

| | |
|---|---|
| **Positive ↔ Negative (Yelp)** | |
| Positive | frozen hot chocolate with peanut butter cups = amazing. i'll be back for some food next time! |
| Negative | frozen hot chocolate with peanut butter ? horrible. i'll stick with the coffee shop next door! |
| Negative | one word: underwhelming. save your money and find the many restaurants in vegas that offers a real experience. |
| Positive | one word: delicious. save room for the best and most authentic indian food in vegas. |
| **Asian ↔ Mexican (Yelp)** | |
| Asian | best thai food i've ever had in the us. great duck specials on monday.. best yellow curry fried rice.. |
| Mexican | best mexican food i've ever had in my life. great guacamole on the side.. best carnitas tacos i have ever had.. |
| Mexican | awesome carne asada! try the papa verde with steak! it's delicious and the portions are great! |
| Asian | awesome orange chicken! try the orange chicken with the spicy sauce! it's delicious and the portions are great! |
| **Male ↔ Female (Yelp)** | |
| Male | good food. my wife and i always enjoy coming here for dinner. i recommend india garden. |
| Female | good food. my husband and i always stop by here for lunch. i recommend the veggie burrito. |
| Female | we are regulars here... me n my husband just gorge on these freaking amazing donuts!!!!! loved it |
| Male | we are regulars here... every time we come here she loves the new york style pizza!!!!!! |
| **Positive ↔ Negative (Amazon)** | |
| Positive | i love this game. takes patience and strategy, go online and look for hints and cheats, they help alot. great game!!! |
| Negative | i don't like this game. it takes a lot of time to figure out how to play, and it doesn't work. i would not recommend this game. |
| Negative | i did not like the conflicting historical data. what was real and what was not. i prefer fiction with facts intact. |
| Positive | i enjoyed the historical references. what a great read and i loved it. i highly recommend this book. |
| **Movies ↔ Books (Amazon)** | |
| Movies | very good movie with outstanding special effects i recommend this to all shi fi lovers. good acting great plot lots of action |
| Books | very good book with outstanding character development i recommend this to all the readers. good job great plot twists and turns |
| Books | definitely not a stone barrington book, but a story told that keeps you wanting more. great read! |
| Movies | definitely not a film noir, but a story that keeps you on the edge of your seat. great acting and a great story. |
| **Clothing ↔ Electronics (Amazon)** | |
| Clothing | got this cause it said it would help with tennis elbow and guess what my tennis elbow still bothering me |
| Electronics | got this cause it said it would help with windows xp and guess what my windows xp still crashed |
| Electronics | i have no choice. this is the only black ink that works with my printer. |
| Clothing | i have no choice. this is the only black color that works with my dress. |

Table 8: Example re-writes by our model on the *FYelp* and *Amazon* datasets when controlling a single attribute. The first line in every box is the pre-specified input with its attribute on the left, and the subsequent line is our model's re-write conditioned on a different attribute value.

| Sentiment | Category | Input / Generations |
|---|---|---|
| | | **Amazon** |
| **Positive** | **Movies** | **exciting new show. john malkovich is superb as always. great supporting cast. hope it survives beyond season 1** |
| Positive | Books | exciting new book. john grisham is one of the best. great read. hope he continues to write more. |
| Negative | Books | nothing new. john grisham is not as good as his first book. not a good read. |
| Positive | Clothing | awesome new watch. fits perfectly. great price. great quality. hope it lasts for a long time. |
| Negative | Clothing | horrible. the color is not as pictured. not what i expected. it is not a good quality. |
| Positive | Electronics | works great. the price is unbeatable. great price. great price. hope it lasts for a long time. |
| Negative | Electronics | worthless. the picture is not as clear as the picture. not sure why it is not compatible with the samsung galaxy s2. |
| Positive | Movies | exciting new show. john goodman is great as always. great supporting cast. hope it continues to end. |
| Negative | Movies | horrible. the acting is terrible. not worth the time. it's not worth the time. |
| Positive | Music | awesome new album. john mayer is one of the best. great album. hope he continues to release this album. |
| Negative | Music | horrible. the songs are not as good as the original. not worth the price. |
| | | **Yelp** |
| **Negative** | **Dessert** | **the bread here is crummy, half baked and stale even when "fresh." i won't be back.** |
| Positive | American | the burgers here are juicy, juicy and full of flavor! i highly recommend this place. |
| Negative | American | the bread here is stale, dry and over cooked even though the bread is hard. i won't be back. |
| Positive | Asian | the sushi here is fresh, tasty and even better than the last. i highly recommend this place. |
| Negative | Asian | the noodles here are dry, dry and over cooked even though they are supposed to be "fresh." i won't be back. |
| Positive | Bar | the pizza here is delicious, thin crust and even better cheese (in my opinion). i highly recommend it. |
| Negative | Bar | the pizza here is bland, thin crust and even worse than the pizza, so i won't be back. |
| Positive | Dessert | the ice cream here is delicious, soft and fluffy with all the toppings you want. i highly recommend it. |
| Negative | Dessert | the bread here is stale, stale and old when you ask for a "fresh" sandwich. i won't be back. |
| Positive | Mexican | the tacos here are delicious, full of flavor and even better hot sauce. i highly recommend this place. |
| Negative | Mexican | the beans here are dry, dry and over cooked even though they are supposed to be "fresh." i won't be back. |

Table 9: Demonstrations of our model's ability to control multiple attributes simultaneously on the *Amazon* dataset (top) and *FYelp* dataset (bottom). The first two columns indicate the combination of attributes that are being controlled, with the first row indicating a pre-specified input

| **Relaxed → Annoyed** | |
| --- | --- |
| Relaxed | Wow! Sitting on my sister's patio, a glass of wine, and glorious music! Hmmmm! |
| Annoyed | Wow! Sitting on my sister's patio, a glass of wine, and the neighbors are out! Geez! |
| Relaxed | Nothing like a few 🍷 🍸 after a long productive day!! |
| Annoyed | Nothing like a flat tire 😠 after a long day at work!! |
| Relaxed | I decided it was time for a little chill time with my people tonight, I Missed them! Plus I need a break. 😌💅 🍷 🍹🎵 |
| Annoyed | I thought I was sleeping for a little while at the end of the week, I'm done! Plus I need a nap. 😠 ♀ 😠😠😠😠 |
| Relaxed | Rain = Sleepy! Love the sound of rain on my tin roof. 🌧💧 FYI: Tomorrow is FRIDAY! |
| Annoyed | Rain = Mad! The sound of rain on my tin roof. 🌧☂ Rain is overrated! |
| Relaxed | Yay!! A quick pedicure before I pick up the little angel from school! 😇 |
| Annoyed | Yay!! A week before I pick up the little bastard from school! FML |
| Relaxed | Had an amazing day driving around. The sea, the woods, just great. 👍 |
| Annoyed | Had an amazing day driving around. The weather, the roads are delayed, and the traffic is closed. 😞 |
| Relaxed | Happiness is watching movie in a rainy day cuddling with your kids in a cozy bed 😊 |
| Annoyed | Yet again watching tv in a rainy day cuddling with my kids in a cozy house 😑 |
| Relaxed | Love sitting in the tree listening to the woods wake up!! Last day of 6 day 😊 |
| Annoyed | Love sitting in the tree listening to the wake up call!! Last day of my holiday was just ruined 😠 |
| Relaxed | A cup of hot creamy #coffee is all i need 😊 #relaxed #focused |
| Annoyed | A cup of hot coffee is the worst feeling ever 😑😑😑 #overit |
| Relaxed | I am in bed, listening to my music. 50 60 70 2000s..... 😄😄◎◎ |
| Annoyed | I am in bed, trying to sleep. This 50 + degree weather..... 😌😌😌 |
| Relaxed | A nice end to a busy day, dinner & drinks with the hubby 💗🎆🍸🙌 |
| Annoyed | A bad end to a busy day, dinner & drinks are just ruined 👿👿🧁🧁🧁 |
| **Annoyed → Relaxed** | |
| Annoyed | Looks like my shovelling is done. Just broke my shovel. |
| Relaxed | Looks like my adjustable couch is done. Just chilling. |
| Annoyed | Nothing makes me more mad on Thursday get paid and the ATM at back is also broke. Thanks for the great service |
| Relaxed | Nothing better than a good massage on the beach and watching the sunset at the pool. Thanks for the great company |
| Annoyed | Who cares that Golden State won... im so over it... ready for football |
| Relaxed | Out of town by the Sea, and... im so excited... ready for vacation |
| Annoyed | Some people just can't be themself! #actthesamearoundeveryone |
| Relaxed | Having a great time with the bestie! #momanddaughterday #muchneeded |
| Annoyed | When gmail is isn't connecting and you tried to get stuff submitted before midnight |
| Relaxed | When the day is just over and you finally get to enjoy some rays before bed |
| Annoyed | I've never been this mad in my life!!! 😠😠😠😤 |
| Relaxed | I've never been this relaxed in my life!!! 💗💗💗💕 |
| Annoyed | I hate it when I can't get to sleep. Stop it brain!! |
| Relaxed | I love it when I can't get up. Beautiful weather!! |
| Annoyed | When u get one of those phone calls that messes up ur day 😼😠 .. |
| Relaxed | When u get one of those massage chairs that helps ur body 😍💗 |
| Annoyed | Don't u just love it when people don't respond to ur texts 😊😠 |
| Relaxed | Don't u just love it when it's time to pamper yourself 😍😍 |
| Annoyed | 5 of battery and no charger since it just broke a min ago ugh!! 😑 |
| Relaxed | 5 minutes of sun and a bottle of beer now just watching a movie 🎥😄🥰 |

Table 10: Examples of our model's ability to re-write sentences from public social media content when conditioned on information about the feeling expressed by the writer (Relaxed vs Annoyed)

| 65+ → 18-24 | |
| --- | --- |
| 65+ | Reality leaves a lot to the imagination. - John Lennon |
| 18-24 | Reality leaves a lot of memes - John Cena |
| 65+ | Photos are beautiful. It must be unreal in person 😄 |
| 18-24 | Cool pic bro - It must be fab in person! |
| 65+ | where did the time go ? LOVE the toothless smile! |
| 18-24 | lemme show u something 😄 ⬆️ 💯 toothless smile 😄 |
| 65+ | so pretty I feed so many in my yard and just love seeing them everyday. |
| 18-24 | so pretty I wanna tag in my family and just love them everyday |
| 65+ | Just where are you. I am loving the pictures and just a bit envious. |
| 18-24 | Just wanna be you I'm loving the pictures and just a bit excited |
| 65+ | Yes, so true! It is incredible what moms learn from daughters and grands! |
| 18-24 | Lmao so true! It's incredible what moms learn from daughters and kids x |
| 65+ | Fantastic picture, Peter you look younger all the time, must be the love! |
| 18-24 | Osm pic Peter you look younger all the time, must be the love 😍 😍 |
| 65+ | What a sweet little girl. One can see the love 💗 between you 2. |
| 18-24 | What a sweet little girl 😍 💯 can see the love 💗 between you 2 |
| 65+ | Congratulation young man. I am very proud of you, keep up the good work. |
| 18-24 | Congratulation sis I'm very proud of you keep up the good work |
| 65+ | So good to hear you are doing well. Love seeing pictures of your precious granddaughter 💗 |
| 18-24 | So good to hear you're doing well 💗 seeing pictures of your precious baby 💗 |
| **18-24 → 65+** | |
| 18-24 | Yeah ight bitch. We gon see who get the last laugh 😏 |
| 65+ | These ignorant idiots. We might see who get the last laugh!!!!! |
| 18-24 | Y'all are cute tho and I'm happy for you guys 💗 |
| 65+ | Hi, you are cute folks and I am happy for you guys. |
| 18-24 | 😄 😄 i dont love boys 😄 😄 they're so urgh 😫 |
| 65+ | What a lovely group of boys! They are so fortunate to have an exceptional faces. |

Table 11: Model re-writes of sentences from public social media content when conditioned on the age-group of the writer (18-24 vs 65+)

| **Positive ↔ Negative (Yelp)** | |
|---|---|
| Positive | happy to find this hidden gem near my office. great food and best of all, fast delivery. |
| Negative | the restaurant near my office was such a dump. late delivery and gross, cold food. |
| Negative | omfg no. i ordered baklava and they nuked it on a styrofoam dish for me. the insides were still cold and it tasted like cancer :/ |
| Positive | yes! i ordered baklava and it was the perfect temperature on a fancy plate. the inside was great and it tasted excellent. |
| **Mexican ↔ Asian (Yelp)** | |
| Mexican | just awful. so called'asada burrito' was cold and bland... just a lump of plain carnitas and some iceberg lettuce in a cheap tortilla. reminiscent of taco bell in the 90s but worse. |
| Asian | just awful. so called spring roll was cold and bland....jump a lump of meat and vegetables in an egg noodle wrapper. reminded me of frozen chinese food but worse. |
| **Asian ↔ American (Yelp)** | |
| Asian | my new favorite curry house! definitely coming back. chicken katsu and takoyaki is soooo goood! cant wait to try the pork katsu. |
| American | my new favorite american bistro house! we are definitely coming back. fried chicken and waffles is soooo good. cant wait to try the new bbq rib sandwich. |
| **Mexican ↔ Dessert (Yelp)** | |
| Mexican | tacos were delicious. tried the carne asad, bbq pork, and chorizo. came with onion and cilantro topping and a house made choice of mild or hot salsa. |
| Dessert | cheesecake was delicious. tried the strawberry flavored one with chocolate drizzle. came with a fresh cherry on top and a house made choice of iced or hot coffee |
| **Positive ↔ Negative (Amazon)** | |
| Negative | too sci fi for me. characters not realistic. situation absurd. i abandoned it halfway through, unusual for me. simply not my taste in mysteries. |
| Positive | i love how sci fi this was! the characters are relatable, and the plot was great. i just had to finish it in one sitting, the story got me hooked. this is has to be one of my favorite mysteries. |
| Positive | my mom love this case for her i-pod. she uses it a lot and she is one satisfied customer. she would recommended it. |
| Negative | my mom initially liked this case for her i-pod. she used it for a while and it broke. since it is not solid she would not recommend it. |
| **Clothing ↔ Books (Amazon)** | |
| Clothing | nice suit but i wear a size 8 and ordered at 12 and it was just a bit too small. |
| Books | great book but i can't read small text with my bad eyesight, unfortunately, this one happened to be printed rather small. |
| **Books ↔ Clothing (Amazon)** | |
| Books | great book about the realities of the american dream. well written with great character development. i would definitely recommend this book. |
| Clothing | great dress with american flags printed on it. well made with great materials. i would definitely recommend this dress. |
| **Books ↔ Music (Amazon)** | |
| Books | i loved the book an can't wait to read the sequel! i couldn't put the book down because the plot and characters were so interesting. |
| Music | i loved the music and can not wait for the next album! i couldn't stop listening because the music was so interesting. |

Table 12: Examples of human edits from our *FYelp* and *Amazon* datasets. The first line in every box was the input presented to a crowd worker followed by their corresponding edit with the specified attribute.

| Accuracy | Self-BLEU | Input / Swap |
|---|---|---|
| | | **not a fan. food is not the best and the service was terrible the two times i have visited.** |
| 78.7% | 73.8 | great little place. food is great and the service was great the two times i have visited. |
| 83.5% | 51.5 | best food in town. food is great and the service was the best two times i have visited. |
| 92.8% | 27.7 | best thai food in town. the food is great and the service is excellent as always. |
| 96.8% | 13.1 | best chinese food in town. great service and the food is the best i have had in a long time. |
| | | **overpriced specialty food. also very crowded. service is slow. would not recommend at any time.** |
| 78.7% | 73.8 | great homemade food. also very crowded. service is fast. would recommend at least once. |
| 83.5% | 51.5 | great specialty food. also very crowded. service is friendly. would recommend any time at the time. |
| 92.8% | 27.7 | great variety of food. also very friendly staff. good service. would recommend at least once a week. |
| 96.8% | 13.1 | great tasting food. very friendly staff. definitely recommend this place for a quick bite. |

Table 13: Examples of controlling sentiment with different model checkpoints that exhibit different trade-offs between content preservation (self-BLEU) and attribute control (Accuracy). The first line in both examples is the input sentence, and subsequent lines are a model's outputs with decreasing content preservation with rows corresponding to model checkpoints at different epochs during training.

| Positive | Negative | Male | Female | American | Asian | Bar | Dessert | Mexican |
|---|---|---|---|---|---|---|---|---|
| pampered | dishonest | ammo | manicure | primanti | guu | promoter | patisserie | tortas |
| relaxation | fraud | tenant | pedi | bobby | chashu | bouncers | froyo | fundido |
| delightfully | incompetence | bachelor | hubs | flay | izakaya | bouncer | buttercream | arepas |
| complemented | unethical | barbers | bridesmaids | lux | tonkotsu | hakkasan | dunkin | burritos |
| cutest | insulted | wife | bridal | nacho | khao | postino | groomers | tostada |
| plush | audacity | firestone | pedicure | bj | soju | bachi | bakeries | taquitos |
| punctual | confronted | provider | instructors | gown | tonkatsu | brio | bakery | nacho |
| housemade | cockroach | data | mattresses | applebee | banchan | cabanas | custard | fajita |
| precision | crooks | plumber | stylist | burgr | shabu | films | doughnuts | guac |
| masterpiece | disrespect | motor | jacuzzi | chilis | teppanyaki | trader | pastries | refried |
| restored | roaches | contractor | hubby | mesa | gai | hooters | gelato | mexico |
| comprehensive | refunds | hertz | pregnancy | bmw | kbbq | karaoke | cheesecakes | empanadas |
| sublime | shrugged | hvac | bachelorette | denny | pho | irish | donut | tapas |
| made | liars | qualified | husbands | mastro | hotpot | harry | croissants | queso |
| tastefully | rudest | incompetence | barre | cellar | soi | darts | doughnut | cantina |
| treasures | accused | transmission | cutest | bachi | karaage | nightclub | danish | salsas |
| addicting | inconsiderate | summary | lashes | rubbed | omakase | applebee | cheesecake | pollo |
| marvelous | roach | contractors | sephora | flatbread | saigon | whiskey | fritter | asada |
| handsome | rudely | automotive | bf | skins | panang | cereal | macarons | barrio |
| healing | cancellation | audio | boyfriends | grille | cantonese | perform | oreos | barbacoa |

Table 14: Examples of the learned attribute biases for sentiment and restaurant categories on *FYelp*.

