# OpenReview forum: "Multiple-Attribute Text Rewriting"
_ICLR.cc/2019/Conference_

### Official Review · AnonReviewer2 · 2018-10-13
**Impressive experiments, but hard to determine how much is methodologically new here**

**Rating:** 6
**Confidence:** 3

**Review:**

The paper proposes "style transfer" approaches for text rewriting that allow for controllable attributes. For example, given one piece of text (and the conditional attributes associated with the user who generated it, such as their age and gender), these attributes can be changed so as to generate equivalent text in a different style.

This is an interesting application, and somewhat different from "style transfer" approaches that I've seen elsewhere. That being said I'm not particularly expert in the use of such techniques for text data.

The architectural details provided in the paper are quite thin. Other than the starting point, which as I understand adapts machine translation techniques based on denoising autoencoders, the modifications used to apply the technique to the specific datasets used here were hard to follow: basically just a few sentences described at a high level. Maybe to somebody more familiar with these techniques will understand these modifications fully, but to me it was hard to follow whether something methodologically significant had been added to the model, or whether the technique was just a few straightforward modifications to an existing method to adapt it to the task. I'll defer to others for comments on this aspect.

Other than that the example results shown are quite compelling (both qualitatively and quantitatively), and the experiments are fairly detailed.

---

> ### Author Response · Authors · 2018-11-17
> **Thank you for the review & comments**
>
> To make the architecture clearer, we updated the paper and added a paragraph, describing the architecture of the model in the “Implementation” section. That paragraph was previously in the appendix -- we hope inserting it into the main body makes the paper easier to follow.
>
> As for our additions to the model, the methodology we used is similar to previous approaches in unsupervised machine translation, but with two key differences.
>
> First, our approach can handle multiple attributes, while previous approaches usually only consider two different domains (one for the positive reviews, and one for the negative reviews, for instance) and cannot be easily extended to multiple domains as they typically require one encoder and one decoder per domain. Our approach can handle multiple attributes at the same time, including categorical attributes (e.g. Table 9 in the Appendix).
>
> Also, we introduced a pooling operator and we found it to be critical in our experiments. The problem we observed is that without it, the model has a tendency to converge to the “copy mode”, where it simply copy words one by one, without taking the attribute input into consideration. We included a plot in the ablation study (Figure 1) that shows the evolution of the attribute transfer accuracy and the content preservation over training, for different pooling layer configurations. We can see that without the pooling operator, the model directly converges to the “copy mode”, with a self-BLEU close to 90 after only a few epochs. A pooling operator with a window of size 8 not only alleviates this issue, but it also provides intermediate models during training with different trade-offs between content-preservation and attribute transfer.

---

### Official Review · AnonReviewer3 · 2018-11-02
**Good work but better presentation needed**

**Rating:** 6
**Confidence:** 4

**Review:**

This work proposes a new model that controls several factors of variation in textual data where the condition on disentanglement is replaced with a simpler mechanism based on back-translation. It allows control over multiple attributes, and a more fine-grained control on the trade-off between content preservation and change of style with a pooling operator in the latent space.

One of the major arguments is it is unnecessary to have attribute-disentangled latent representations in order to have good style-transferring rewriting. In Table 2, the authors showed that "a classifier that is separately trained on the resulting encoder representations has an easy time recovering the sentiment" when the discriminator during training has been fooled. Is there any difference between the two discriminators/classifiers? If the post-fit classifier on top of the encoder representation can easily predict the correct sentiment, there should be enough signal from the discriminator to adapt the encoder in order to learn a more disentangled representation. On the other hand, this does not answer the question if a "true" disentangled representation would give better performance. The inferior performance from the adversarially learned models could be because of the "entangled" representations.

As the author pointed out, the technical contributions are the pooling operator and the support for multiple attributes since the loss function is the same as that in (Lample et. al 2018). These deserve more elaborated explanation and quantitative comparisons. After all, the title of this work is "multiple-attribute text rewriting". For example, the performance comparison between the proposed how averaged attribute  embeddings and simple concatenation, and the effect of the introduced trade-off using temporal max-pooling.

How important is the denoising autoencoder loss in the loss function (1)? From the training details in the supplementary material, it seems like the autoencoder loss is used as "initialization" to some degree. As pointed out by the authors, the main task is to get fluent, attribute-targeted, and content-preserving rewriting. As long as the "back-translation" gives expected result, it seems not necessary to have "meaningful" or hard "content-preserving" latent representations when the generator is powerful enough.

I think the last and most critical question is what the expected style-transferred rewriting look like. What level or kind of "content-preserving" do we look for? In Table 4, it shows that the BLEU between the input and the referenced human rewriting is only 30.6 which suggest many contents have been modified besides the positive/negative attribute. This can also be seen from the transferred examples. In Table 8, one of the Male example: "good food. my wife and i always enjoy coming here for dinner. i recommend india garden." and the Female transferred rewriting goes as "good food. my husband and i always stop by here for lunch. i recommend the veggie burrito". It's understandable that men and women prefer different types of food even though it is imagination without providing context. But the transfer from "dinner" to "lunch" is kind of questionable. Is it necessary to change the content which is irrelevant to the attributes?


Other issues:
- Towards the end of Section 3, it says that "without back-propagating through the back-translation generation process". Can you elaborate on this and the reason behind this choice?
- What does it mean by "unknown words" in "... with 60k BPE codes, eliminating the presence of unknown words" from Section 4?
- There is no comparison with (Zhang et. al. 2018), which is the "most relevant work".
- In Table 4, what is the difference among the three "Ours" model?
- In Table 4, the perplexity of "Input Copy" is very high compared with generated sentences.
- In Table 7, what does the "attention" refer to?
- In the supplementary material, there are lambda_BT and lambda_AE. But there is only one lambda in the loss function (1).
- Please unify the citation style.

---

> ### Author Response · Authors · 2018-11-17
> **Thank you for your review and raising interesting questions about this work. (part 1)**
>
> “Is there any difference between the two discriminators/classifiers?” - The discriminator and classifier have completely identical architectures - a 3 layer MLP with 128 dimensional hidden layers and LeakyReLU activations (now clarified in the model architecture paragraph in Section 3.3). We used two different terms to describe them since the classifier is fit post-hoc and doesn’t adapt to the encoder representations in a min-max fashion while the discriminator does. Moreover, the classifier is fully trained on the final encoder representations, while the discriminator is “chasing” them without fully training after each and every update of the encoder representations. This is indeed a bit confusing, and we have clarified this in the paper. While a discriminator trained more thoroughly at each iteration might disentangle representations more, our goal was not to look at whether disentangled representations can result in better performance, but whether current training practices actually result in disentangled representations (see responses below as well).
>
> “there should be enough signal from the discriminator to adapt the encoder in order to learn a more disentangled representation.” - This is a valid concern, but the experiments we ran suggest that this does not change the main observation. For instance, we also experimented with larger coefficients of adversarial training of 1.0 and 10.0 (as well as no adversarial training on the other end of the spectrum). While the attribute recovery accuracy drops a little at higher coefficients, it is still much higher than the discriminator accuracy during training. Also, models trained with high adversarial training coefficients have extremely high reconstruction and back-translation losses. Results are presented below, for better formatting please refer to the revised version of our paper.
>
>         Coef        Disc(acc)  Clf(acc)
>         0        &	89.45% &  93.8%
>         0.001 &	85.04% &  92.6%
>         0.01   &	75.47% &  91.3%
>         0.03   &	61.16% &  93.5%
>         0.1     &	57.63% &  94.5%
>         1.0     &	52.75% &  86.1%
>         10      &	51.89% &  85.2%
>
> “On the other hand, this does not answer the question if a "true" disentangled representation would give better performance. The inferior performance from the adversarially learned models could be because of the "entangled" representations.” - We agree completely. Our point is not that disentangled representations would not lead to good performance, but simply that disentanglement doesn't happen in practice with the kind of adversarially trained models typically used for this problem. We have made changes to the writing to make our stance clearer.
>
> “Request for ablation study on pooling and other architectural design choices.” - In addition to the averaged attribute embeddings, we also explored using a separate embedding for each attribute combination in the cross-product of all possible attribute values. We found this to have similar performance to our averaging method. We decided against concatenating embeddings because we use the attribute embedding as the first input token to the decoder, and using a concatenation would mean dividing the embedding size for each attribute value by the number of attributes, to maintain to overall embedding size. This wouldn’t scale as well to settings with many possible attributes. We settled on the attribute embedding averages because of its simplicity.
>
> We have included a plot (Figure 1) that shows the evolution of attribute control (accuracy) and content preservation (BLEU) over the course of training as a function of the pooling kernel width. This demonstrates the latent space pooling operator’s ability to trade off self-BLEU and accuracy - larger kernel widths favor attribute control while smaller ones favor content preservation.
>
> “As long as the "back-translation" gives expected result, it seems not necessary to have "meaningful" or hard "content-preserving" latent representations when the generator is powerful enough.”
> We observed that operating without a DAE objective didn’t work since the model needs to be bootstrapped to be capable of producing outputs that are at least somewhat close to the original input before the back-translation process can take over. At the beginning of training, it is nearly impossible for the model to be able to recover the original input starting from a nearly random sequence of words. But it’s indeed true that later on the back-translation loss is enough: in practice, we in fact removed the DAE objective by progressively decreasing lambda_AE from 1 to 0 over the first 300,000 iterations (c.f. Appendix section), even though we didn’t observe a significant difference compared to simply fixing lambda_AE to 1.

---

> ### Author Response · Authors · 2018-11-17
> **Thank you for your review and raising interesting questions about this work. (part 2)**
>
> “I think the last and most critical question is what the expected style-transferred rewriting look like. What level or kind of "content-preserving" do we look for?” - This is a great question, and a fundamental open research problem which, as far as we know, does not have a clear answer in existing literature. In our paper,  we view this line of research as looking for better ways to generate rewrites of text along certain directions, and exactly the “kind” of what content is being preserved would ideally be one of the “knobs” that a system can control. The phrase “style transfer” is useful to refer to previous work that have adopted it from the image domain, but its framing is a bit narrow for the scope of rewriting types our work addresses. We believe that the trade-off between attribute control and content preservation should depend on two factors 1) the eventual use case of such a system (and style transfer is one use case, but another one would be to obtain more “interesting” and varied generations by augmenting a retrieval system with rewriting capabilities in a controllable way, and 2) the nature of attributes being controlled. Firstly, in contrast to previous work, we present means to control this inherent trade-off in the form of a latent-space pooling operator which can adapted to a particular use case. Secondly, the proposed method is fundamentally one that learns an unsupervised mapping between two or more domains of text, and the nature of the learned mapping will certainly depend on the nature of the domains. For example, it is often possible to map between the positive and negative domains by replacing a few words or small phrases and as a result, we can expect our models to preserve a lot of the input. By contrast, attributes such as one’s age aren’t as “local” and might require rewriting more content to successfully be altered. In that case, the content that is being preserved might be the general structure of the text, its sentiment, etc. To make the trade-off clearer, we have added a figure to the manuscript showing how it varies across training (Fig. 1 in the appendix); we also include illustrations of rewrites at different trade-off levels in Table 13.
>
> “Towards the end of Section 3, it says that "without back-propagating through the back-translation generation process". Can you elaborate on this and the reason behind this choice?” - Back-propagating through the back-translation process would require computing gradients through a sequence of discrete actions since generations are sampled from the decoder. While this may be achieved via policy-gradient methods such as REINFORCE or other approximations like the Gumbel-softmax trick, these have been known to perform very poorly in high dimensional action spaces due to high variance of the gradient estimates. This approach also has the disadvantage of biasing the model towards the degenerate solution of copying the input while ignoring attribute information entirely to satisfy the cycle-consistency objective, since the gradients flow through the entire cycle, which is what we observed in practice.
>
> “What does it mean by "unknown words" in "... with 60k BPE codes, eliminating the presence of unknown words" from Section 4?” - We meant that by using BPE, we can operate without replacing infrequent words with an <unk> token -- we do not have unknown words because these are decomposed into subword units that belong to the BPE dictionary.
>
> “what is the difference among the three "Ours" model?” - These models differ in the choice of hyperparameters (pooling kernel width and back-translation temperature) to demonstrate our model’s ability to control the content preservation vs attribute control trade-off. We have clarified this in the table caption.
>
> “the perplexity of "Input Copy" is very high compared with generated sentences.” - This is true and we believe that this is a consequence of the fact that there is more diversity in the input reviews than in typical generations from ours and other systems. This lack of diversity is typical for models decoding with beam search, which leads to "mode seeking behavior" wherein the output generations contain fragments that occur most frequently in the training set. This results in the pre-trained LM assigning high likelihoods to these samples.
>
> “what does the "attention" refer to?” - The row in Table 7 that corresponds to "-attention" refers to a model that was trained without an attention mechanism in a vanilla sequence-to-sequence fashion, using the last hidden state of the encoder by concatenating it to the word embeddings at every time step of the decoder.
>
> “In the supplementary material, there are lambda_BT and lambda_AE. But there is only one lambda in the loss function (1).” -Thank you for spotting this typo. We fixed this in the revised version of the paper.

---

### Official Review · AnonReviewer1 · 2018-11-05
**This paper presents a model for text rewriting for multiple attributes.**

**Rating:** 7
**Confidence:** 3

**Review:**

This paper presents a model for text rewriting for multiple attributes, for example gender and sentiment, or age and sentiment. The contributions and strengths of the paper are as follows.

* Problem Definition
An important contribution is the new problem definition of multiple attributes for style transfer. While previous research has looked at single attributes for rewriting, "sentiment" for example, one could imagine controlling more than one attribute at a time.

* Dataset Augmentation
To do the multiple attribute style transfer, they needed a dataset with multiple attributes. They augmented the Yelp review dataset from previous related paper to add gender and restaurant category. They also worked with microblog dataset labeled with gender, age group, and annoyed/relaxed. In addition to these attributes, they modified to dataset to include longer reviews and allow a larger vocabulary size. In all, this fuller dataset is more realistic than the previously release dataset.

* Model
The model is basically a denoising autoencoder, a well-known, relatively simple model. However, instead of using an adversarial loss term as done in previous style transfer research, they use a back-translation term in the loss. A justification for this modeling choice is explained in detail, arguing that disentanglement (which is a target of adversarial loss) does not really happen and is not really needed. The results show that the new loss term results in improvements.

* Human Evaluation
In addition to automatic evaluation for fluency (perplexity), content preservation (BLEU score), and attribute control (classification), they ask humans to judge the output for the three criteria. This seems standard for this type of task, but it is still a good contribution.

Overall, this paper presents a simple approach to multi-attribute text rewriting. The positive contributions include a new task definition of controlling multiple attributes, an augmented dataset that is more appropriate for the new task, and a simple but effective model which produces improved results.

---

> ### Author Response · Authors · 2018-11-17
> **Thank you for the review!**
>
> Thank you for your review. We are glad to see that you liked the paper and it's contributions.

---

### Public Comment · (anonymous) · 2018-11-05
**Empirical comparison to (Hu el al., 2017)**

Thanks for the interesting work. It'd be nice to see an empirical comparison of this work to (Hu el al., 2017) which has released code here: https://github.com/asyml/texar/tree/master/examples/text_style_transfer. Based on my experience, (Hu el al., 2017) is usually a strong baseline on many datasets.

---

> ### Public Comment · (anonymous) · 2018-11-06
> **missing references**
>
> There are also some relevant works that are missing in the references such as:
> Unsupervised Text Style Transfer using Language Models as Discriminators by Yang etc al.

---

> ### Author Response · Authors · 2018-11-17
> **Results of the comparison**
>
> Thank you for your comment. We’ve added a comparison with Hu et al., 2017 in the revised paper using the code you mentioned. We found that this model obtained a good accuracy / BLEU score, but with pretty high perplexity. We’ve also added a reference to Yang et al in the related work section, thank you for pointing this out.

---

### Public Comment · (anonymous) · 2018-11-14
**Convergence question in equation. (1)**

Thanks for the interesting work. The results look amazing. I have a question about the loss function (Equation. 1). The loss function only consists of the reconstruction loss and another type reconstruction loss related to the back-translation, and there is no adversarial loss or classification loss to regularize the generated styles. How do you guarantee the generated sentences have correct styles?

I can imagine that there is a local minimum of Equation. 1, where the decoders completely ignores the input style embedding and directly copy the input sentence.  In this case, no matter which style you used, the input and output are the same, and the loss is zero. I'm wondering how do you prevent this situation happens?

Looking forward to seeing the answers!

---

> ### Comment · AnonReviewer3 · 2018-11-14
> **the denoising matters here**
>
> The first part of the loss function (1), the denoising auto-encoder part, would help prevent the situation described. Since the input would have noise added, the simple copy operation can not be learned directly. But I still prefer the authors would give some discussions and maybe quantitative results regarding this.

---

> > ### Public Comment · (anonymous) · 2018-11-15
> > **The denoising auto-encoder part still couldn't guarantee the styles**
> >
> > Thanks for the comment! Yes, the denoising auto-encoder part could prevent the directly copying. However, it still couldn't guarantee the generated styles. If the auto-encoder totally ignores the style embedding and only learns to reconstruct the input sentences (even with noises), the equation. (1) is still converged.  Hope the authors would discuss this issue.

---

> > > ### Author Response · Authors · 2018-11-17
> > > **Regarding the convergence of the model**
> > >
> > > Thank you for your question. As AnonReviewer3 mentioned, simply copying input sentences wouldn’t satisfy the auto-encoding part of equation (1), as noise has been added to sentences. However, it would indeed satisfy the back-translation loss.
> > >
> > > The idea of denoising here is that by removing random words from a sentence, we hope to remove words that are required to infer the style.
> > > For instance, if the input sentence is: “this place is awful”
> > > and that the noised sentence becomes: “this place is <BLANK>”,
> > > the model will be trained to recover “this place is awful”
> > > from: (“this place is <BLANK>”, ATTRIBUTE=NEGATIVE)
> > >
> > > Since there might be a lot of occurrences of “this place is amazing” in the dataset, the model will have to learn to consider the provided attribute in order to give a high probability to “awful” without penalizing the perplexity on the positive reviews.
> > >
> > > The general argument is that the decoder needs to learn to use the attribute information whenever the input to the system is very noisy. This applies as well when inputs come from the back-translation process. Noisy inputs are produced in the back-translation process at the beginning of training when the model is insufficiently trained and does not generate well, and when generations are produced at high temperature. When using high softmax temperatures, the model tends to exhibit lower content preservation and higher attribute transfer since the generated samples are very noisy and it is therefore more difficult to recover the original input in the back-translation process while the decoder is forced to better leverage the attribute information.

---

### Meta-Review · Area_Chair1 · 2018-12-13
**A simple and effective approach to style transfer based on recent developments in unsupervised NMT**

**Confidence:** 4
**Recommendation:** Accept (Poster)

**Metareview:**

The paper shows how techniques introduced in the context of unsupervised machine translation can be used to build a style transfer methods.

Pros:

-  The approach is simple and questions assumptions made by previous style transfer methods (specifically, they show that we do not need to specifically enforce disentanglement).

-  The evaluation is thorough and shows benefits of the proposed method

-  Multi-attribute style transfer is introduced and benchmarks are created

-  Given the success of unsupervised NMT, it makes a lot of sense to see if it can be applied to the style transfer problem

Cons:

- Technical novelty is limited

- Some findings may be somewhat trivial (e.g., we already know that offline classifiers are stronger than the adversarials, e.g., see Elazar and Goldberg, EMNLP 2018).